# Multi-Scenario Simulation of Land Use Changes with Ecosystem Service Value in the Yellow River Basin

**Yuanyuan Lou** [1], **Dan Yang** [1], **Pengyan Zhang** [1,2,3,*], **Ying Zhang** [1], **Meiling Song** [1], **Yicheng Huang** [1] and **Wenlong Jing** [4]

1  College of Geography and Environmental Science, Henan University, Kaifeng 475004, China; louyuanyuannet@henu.edu.cn (Y.L.); yangdan219@henu.edu.cn (D.Y.); zhangYing@henu.edu.cn (Y.Z.); songmeiling@henu.edu.cn (M.S.); huangyicheng@henu.edu.cn (Y.H.)
2  Key Laboratory of Geospatial Technology for the Middle and Lower Yellow River Regions, Ministry of Education, Henan University, Kaifeng 475004, China
3  Regional Planning and Development Center, Henan University, Kaifeng 475004, China
4  Guangdong Province Engineering Laboratory for Geographic Spatio-temporal Big Data, Key Laboratory of Guangdong for Utilization of Remote Sensing and Geographical Information System, Guangdong Open Laboratory of Geospatial Information Technology and Application, Guangzhou Institute of Geography, Guangdong Academy of Sciences, Guangzhou 510070, China; jingwl@lreis.ac.cn
*  Correspondence: pengyanzh@henu.edu.cn

**Abstract:** Land use change plays a crucial role in global environmental change. Understanding the mode and land use change procedure is conducive to improving the quality of the global eco-environment and promoting the harmonized development of human–land relationships. Large river basins play an important role in areal socioeconomic development. The Yellow River Basin (YRB) is an important ecological protective screen, economic zone, and major grain producing area in China, which faces challenges with respect to ecological degradation and water and sediment management. Simulating the alterations in ecosystem service value (ESV) owing to land use change in the YRB under multiple scenarios is of great importance to guaranteeing the ecological security of the basin and improve the regional ESV. According to the land use data of 1990, 2000, 2010, and 2018, the alterations in the land use and ESV in the YRB over the past 30 years were calculated and analyzed on the basis of six land use types: cultivated land, forestland, grassland, water area, built-up land, and unused land. The patch-generating land use simulation (PLUS) model was used to simulate the land use change in the study area under three scenarios (natural development, cultivated land protection, and ecological protection in 2026); estimate the ESV under each scenario; and conduct a comparative analysis. We found that the land use area in the YRB changed significantly during the study period. The ESV of the YRB has slowly increased by ~USD 15 billion over the past 30 years. The ESV obtained under the ecological protection scenario is the highest. The simulation of the YRB's future land use change, and comparison and analysis of the ESV under different scenarios, provide guidance and a scientific basis for promoting ecological conservation and high-quality development of river basins worldwide.

**Keywords:** spatiotemporal analysis; PLUS model; scenario simulation; ecosystem service; yellow river basin

## 1. Introduction

Ecosystem services (ESs) provide benefits that humans derive directly or indirectly from the ecosystem [1,2], and are the foundation of human survival and development [3]. The ecosystem service value (ESV) is an economic measure of the social products and services the ecosystem provides to mankind [4–6]. It is used to evaluate the contribution of ecosystems to sustainable well-being [7]. ESV is a tool that used to appraise the strength of ES, and quantitatively describes the characteristics of spatiotemporal changes and conducts

scientific evaluation, and has important implications for the ecological protection of the region and the enactment of rational policy measures [8]. Land use change is a direct manifestation of human activity and interaction with the environment and has a significant effect on the environment [9,10]. Land use changes convert the structure and function of ecosystems by affecting the soil characteristics and intensity of human activities, thus threatening biodiversity and affecting the ecosystem service capacity and value [11,12]. The equitable use and valid configuration of land are directly related to the sustainable socioeconomic development [13,14]. Currently, research on the ecological impact of land use change has gradually increased, especially with regard to problems related to the ESV.

The ESV is an important aspect of ecological study [15]. Scholars combine the ESV with land use change [16], ecological sensitivity [17], and ecological carrying capacity [18]; use the benefit transfer, the InVEST model [19], ecological footprint [20], and other methods to explore and study the ESV; and take notice of the impact of land use change on the ESV. Currently, the most used indirect evaluation method is the equivalent coefficient method, i.e., the product of the ESV per unit area and land use area [21]. Costanza et al. [22] established the global ESV equivalent factor table in 1997, laying a theoretical foundation for relevant research. Subsequently, researchers have studied the changes in the ESV at different scales (global [23], national [24], and coastal [25]). In accordance with Costanza et al. [22], Xie et al. [26] formulated an equivalent table of the ESV per unit area for China's terrestrial ecosystem. Subsequently, Chinese researchers have used land use data at the national [27], provincial [28], city [29], basin [30], and urban agglomeration scale [31] to determine the time scale of the ESV and land use [32], spatiotemporal distribution of the ESV [33], influence factors [34], and driving forces [35]. Several researchers have used the grey prediction (1,1) model (GM) [36], CA-Markov [37], CLUE-S [38], and FLUS [39] to study the impact on future land use change on the regional ESV and optimize the land use structure based on the ESV. However, study areas mainly included mountains [40], plateaus [41], lakes [42], national parks [43], and ecologically fragile or small-scale areas [44], but there is a lack of multi-scenario prediction simulations of large-scale, complex future land use patterns such as river basins. Therefore, in this study, the PLUS model was used to simulate the change in the land use pattern in the YRB under three different scenarios, and reveal the patch-level evolution of the simulated landscape type such that it better conforms to the actual complex landscape pattern in the river basin. The PLUS model further improves the mining space conversion rules. Therefore, it is more flexible in handling polytype of land use patch changes, and can better simulate the changes of multiple types of land use patch levels.

Development construction of river basins have important effects on sustainable regional development. The YRB is an important ecological barrier, major grain-producing area, and economic zone in China. Whether from the cultural, economic, or ecological perspective, it has always occupied an important position in China. However, the YRB is highly populated. Based on the rapid development of urbanization and extensive human activities, ecological degradation in the YRB is extensive and the annual soil erosion problem is severe. Coupled with water pollution, the ecological security of the YRB is threatened [45]. Currently, preventing ecological degradation is the primary task, with respect to high-quality development in the YRB [46]. According to the characteristics of land use change in the YRB, PLUS was used to set up multiple scenarios to simulate the land use change pattern of future river basin, as well as the change in the ESV. Our specific objectives are: (1) to explore the spatial-temporal changes of land use and ESV in the YRB from 1990 to 2018; (2) using the PLUS model, to predict the land use distribution under three scenarios of natural development, ecological protection and cultivated land protection in the YRB in 2026; and (3) to evaluate the ESV under three future scenarios. To provide scientific and theoretical basis for the ecological protection and high-quality development of the YRB and the strategic goal of regional social and economic sustainable development, we can explore the methods to improve the ESV in the river basin (Figure 1). At the same time, the research results are expected to provide some references for future

land use planning and management, ecological security patterns, and ecological civilization construction in the YRB and other large river basins.

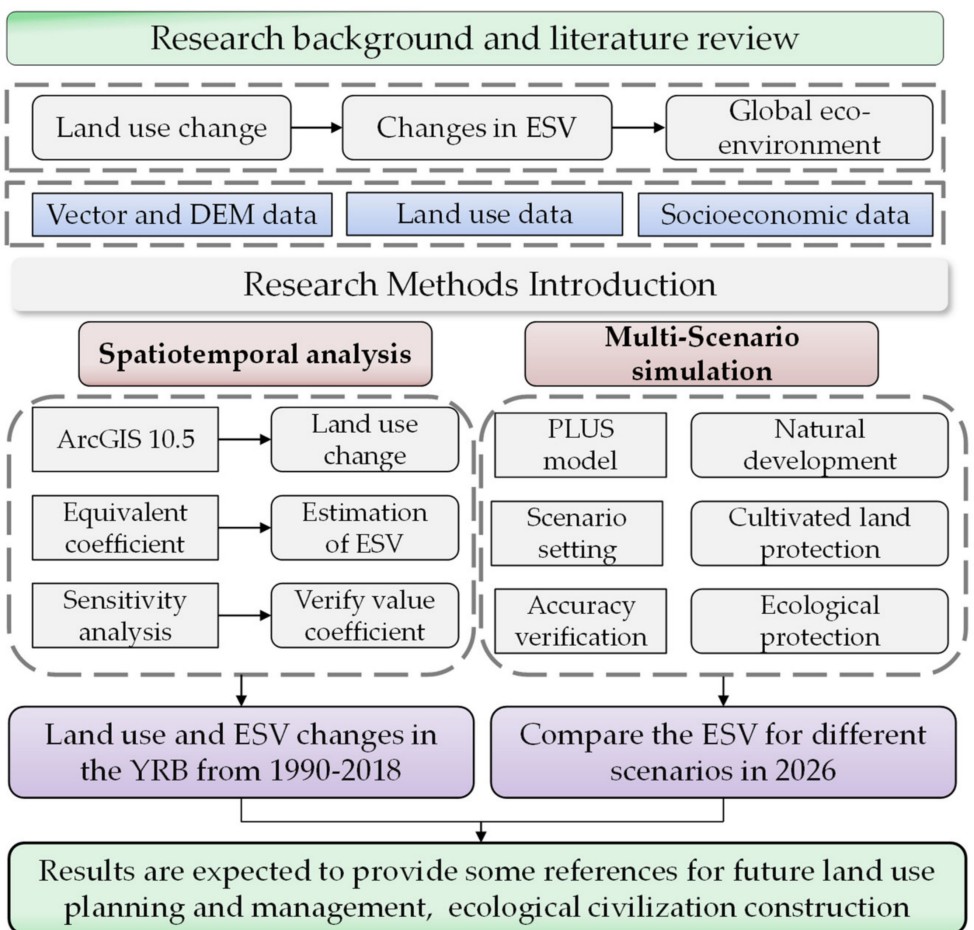

**Figure 1.** Flow chart of the work progress.

## 2. Materials and Methods

### 2.1. Study Area

YRB is vast and has a developed water system. The total length and water depth of the main stream are 5464 km and 4480 m, respectively. The upper reaches include the area from the source of the Yellow River to Hekou Town of the Inner Mongolia Autonomous Region, the middle reaches include the area from Hekou Town to Taohua Valley in Zhengzhou and the lower reaches range from Peach Blossom Valley to the Bohai Sea (Figure 2). Referring to previous studies, 73 cities (prefecture-level cities, prefectures, and alliances) flowing through the provinces were divided into the YRB [47]. The YRB spans the three geographical steps of China, with high and low terrain in the west and east, respectively. It is an important ecological corridor connecting the Qinghai–Tibet Plateau, Loess Plateau, and Yellow–Huai–Hai Plain [48]. It is an important ecological region in China [49].

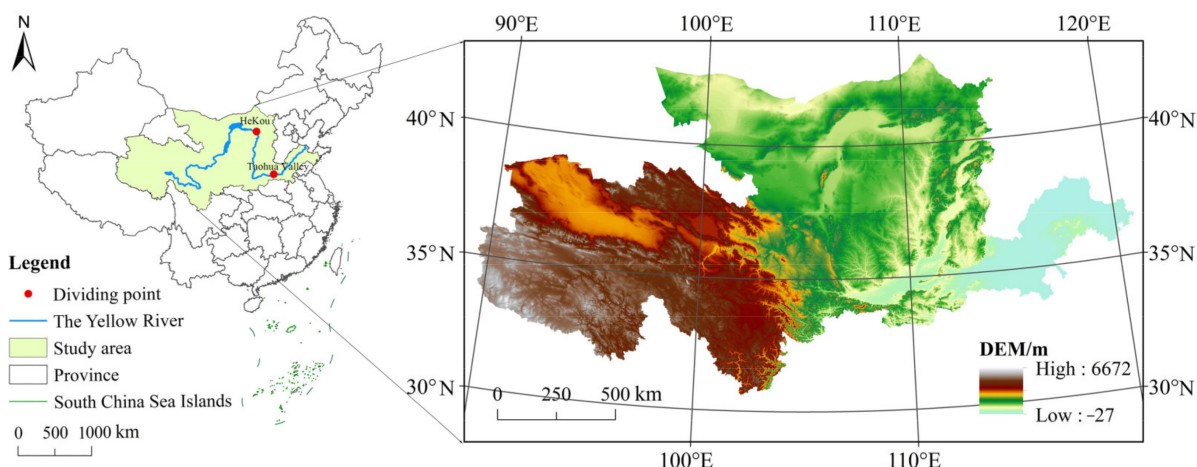

**Figure 2.** Location of the study area.

*2.2. Data Sources*

The study mainly used vector data, digital elevation model (DEM) data, and land use and socioeconomic data for the YRB. The vector data, digital elevation model (DEM) data (spatial resolution of 1 km) and land use data for 1990, 2000, 2010, and 2018 (accuracy is 1 km × 1 km) were obtained from the Resource and Environmental Science Data Center of the Chinese Academy of Sciences (http://www.resdc.cn/, accessed on 1 September 2021). Based on the research needs, the land use types were reclassified into six categories According to the "China Land Use/Land Cover Remote Sensing Monitoring Data Classification System": cultivated land, forestland, grassland, water area, built-up land, and unused land. Socioeconomic data, including the single yields and purchase prices of the main food crops in the YRB (wheat, corn, and rice), were derived from the Provincial Statistical Yearbook of the YRB, China Agricultural Statistical Yearbook, and China Agricultural Products Price Survey Yearbook (https://data.cnki.net/Yearbook, accessed on 1 September 2021).

*2.3. Methods*

2.3.1. Estimation of the ESV

The actual ESV of the YRB was calculated and corrected. First, the actual value of the currently cultivated land area was modified; wheat, corn, and rice, i.e., the main grain crops in the YRB, were selected based on the grain planting areas of 1990, 2000, 2010, and 2018; and the economic output value was determined. According to the study on by Xie et al. [50], the value of the single equivalent factor is ~1/7 of the actual value of the average grain yield per unit area. This equation is as follows [50]:

$$VC = \frac{1}{7}\sum_{i=1}^{n} \frac{m_i}{M_i},\qquad(1)$$

where $VC$ represents the value of a single equivalent factor, $i$ represents the main grain crops, $n$ represents the total category of the main grain crops, $m_i$ represents the economic output value of the $i$th grain crop, and $M_i$ represents the total sown area of the $i$th grain crop.

Subsequently, the individual equivalent factor value in the YRB ESs during the study period was determined. With reference to the equivalent value scale propounded by Xie et al. [50], the ESV per unit area can be calculated for each land use type [50]:

$$ESV = \sum A_j \times VC_j \qquad(2)$$

where $A_j$ means the area of the class $j$ land use type and $VC_j$ expresses the ESV per unit area of the class $j$ land use type.

In reference to the Provincial Statistical Yearbook of the YRB and China Agricultural Product Price Survey Yearbook, we calculated the market price of wheat, corn, and rice per unit yield in the YRB from 1990 to 2018 to be 0.36 USD/kg. The calculation was based on an ESV of cultivated land per unit area of 1/7 of the market economic value of average grain per unit yield and table of ESV coefficients per unit area for various land use types in the YRB (Table 1).

**Table 1.** ESV coefficients for the YRB.

| | | Ecosystem Service Value (USD hm$^{-2}$ a$^{-1}$) | | | | |
|---|---|---|---|---|---|---|
| | | Ecosystem-Type | | | | |
| | | Forest Land | Grass Land | Cultivated Land | Water Area | Unused Land |
| Provisioning Services | Food production | 83.49 | 108.79 | 253.00 | 225.17 | 5.06 |
| | Raw materials production | 753.94 | 91.08 | 98.67 | 149.27 | 10.12 |
| | Subtotal | 837.43 | 199.87 | 351.67 | 374.44 | 15.18 |
| Regulating Services | Gas regulatin | 1092.96 | 379.50 | 182.16 | 738.76 | 15.18 |
| | Climatic regulation | 1029.71 | 394.68 | 245.41 | 3949.33 | 32.89 |
| | Hydrologic regulatin | 1034.77 | 384.56 | 194.81 | 8149.12 | 17.71 |
| | Waste disposal | 435.16 | 333.96 | 351.67 | 7400.24 | 65.78 |
| | Subtotal | 3592.60 | 1492.70 | 974.05 | 20237.45 | 131.56 |
| Supporting Services | Soil retention | 1017.06 | 566.72 | 371.91 | 607.20 | 43.01 |
| | Maintain biological diversity | 1141.03 | 473.11 | 258.06 | 1801.36 | 101.20 |
| | Subtotal | 2158.09 | 1039.83 | 629.97 | 2408.56 | 144.21 |
| Cultural Services | Aesthetic landscape | 526.24 | 220.11 | 43.01 | 2309.89 | 60.72 |
| | Subtotal | 526.24 | 220.11 | 43.01 | 2309.89 | 60.72 |
| | Amount to | 7114.35 | 2952.51 | 1998.70 | 25330.33 | 351.67 |

2.3.2. Sensitivity Analysis

In this study, the elastic coefficient in economics was used for the sensitivity index (coefficient of sensitivity, CS) [51] analysis to verify the accuracy of the effect of the value coefficient (VC) on the *ESV* value. When *CS* > 1, the *ESV* is elastic to the *VC*, When *CS* < 1, the ESV is inelastic to the *VC*, i.e., a 1% *VC* change will cause the *ESV* to change by less than 1%; thus, the research results are reliable.

$$CS = \left| \frac{(ESV_q - ESV_p)/ESV_p}{(VC_{qj} - VC_{pj})/VC_{pj}} \right|, \tag{3}$$

where *p* and *q* represent the coefficient before and after the adjustment, respectively.

2.3.3. PLUS Model

PLUS was developed at the HPSCIL@CUG Laboratory to uncover potential drivers and their contributions to changes [52]. A method combining the Cellular Automata (CA) model with a patch-generation simulation strategy. That basic equation is as follows [52]:

$$OP_{i,j}^{d=1,t} = \begin{cases} P_{i,j}^{d=1} \times (r \times \mu_j) \times H_j^t \ if \ \Omega_{i,j}^t = 0 \ and \ r < P_{i,j}^{d=1} \\ P_{i,j}^{d=1} \times \Omega_{i,j}^t \times H_j^t \ all \ others \end{cases}, \tag{4}$$

where *r* is a random value between 0 and 1; $\mu_j$ is the threshold of new land use patches with land use type *j* determined by the model user [52].

If the new land use type wins a competition round, the decreasing threshold τ is used to evaluate the candidate land use type c selected by the roulette wheel. The formula is as follows [52]:

$$If \sum_{j=1}^{N} \left| G_c^{t-1} \right| - \sum_{j=1}^{N} \left| G_c^t \right| < Step\ Then, l = l + 1, \tag{5}$$

$$\begin{cases} Change\ P_{i,c}^{d=1} > \tau\ and\ T_{j,c} = 1 \\ No\ change\ P_{i,c}^{d=1} \leq \tau\ or\ T_{j,c} = 0 \end{cases} \tag{6}$$

where *Step* is the step of PLUS model, which is used to approximate the demand of land use; δ is the attenuation threshold τ, ranging from 0 to 1, set by an expert; *r*1 is a random value with normal distribution, with an average value of 1, ranging from 0 to 2; *l* is the decay steps. Parameter $TM_{j,c}$ is the transfer matrices that define land use type [53].

### 2.3.4. Scenario Settings

The socioeconomic and natural environments of the YRB are relatively complex, with different regional development plans and complex and diverse landscape patterns. Therefore, based on the different conservation goals and potential interference scenarios in the YRB, three scenarios were analyzed in this research:

(1) Natural development scenario. Based on the land use change in the YRB from 1990 to 2018, current natural development mode of urbanization, and land use transfer rate, this scenario does not restrict other land types in the transfer matrix without considering the impact of policy planning, except that the water area cannot be converted.

(2) Cultivated land protection scenario. Cultivated land security is the basis for ensuring national security. The lower reaches of YRB are an important grain producing area in China. According to the requirements of ecological protection and high-quality development in the YRB, China will continue to consolidate its position as the country's main grain producing region by 2030. This scenario fully considers the need for cultivated land protection. Based on the natural development scenario, it is restricted that cultivated land cannot be converted to other land use types, and other land use types can be converted to cultivated land.

(3) Ecological protection scenario. This scenario aims to be guided by ecological protection and high-quality development in the YRB, adhere to ecological priority, green development, protect the ecological, and vigorously implement forest and grass protection. Based on the natural development scenario, this scenario controls the conversion of forestland, grassland, and water areas to built-up land, cultivated land and unused land, and other land types can flow to forestland, grassland and water area.

## 3. Results

### 3.1. The Estimation and Variation of the ESV

### 3.1.1. General Characteristics of Land Use Change

The evolution of the land use pattern is a vital source of information for understanding ecological change and the impacts of human activities [54]. By collecting and processing the land use data of the YRB for 1990, 2000, 2010 and 2018 and reclassifying various categories at the first level, the distribution maps of cultivated land, forestland, grassland, water area, built-up land, and unused land in the YRB were obtained (Figure 3). According to land use distribution in the YRB study period, the overall land use characteristics in the YRB are as follows: the west is dominated by grassland and forestland; the east is dominated by cultivated and built-up land; the water area is sporadically distributed; and most of the unused land is distributed in the northwest. From the perspective of the upper, middle, and lower reaches, the difference in the distribution of land use types is more notable: that upper reaches are dominated by unused land, grassland, and forestland; the lower reaches are dominated by cultivated and built-up land; and the middle reaches are distributed in the whole country.

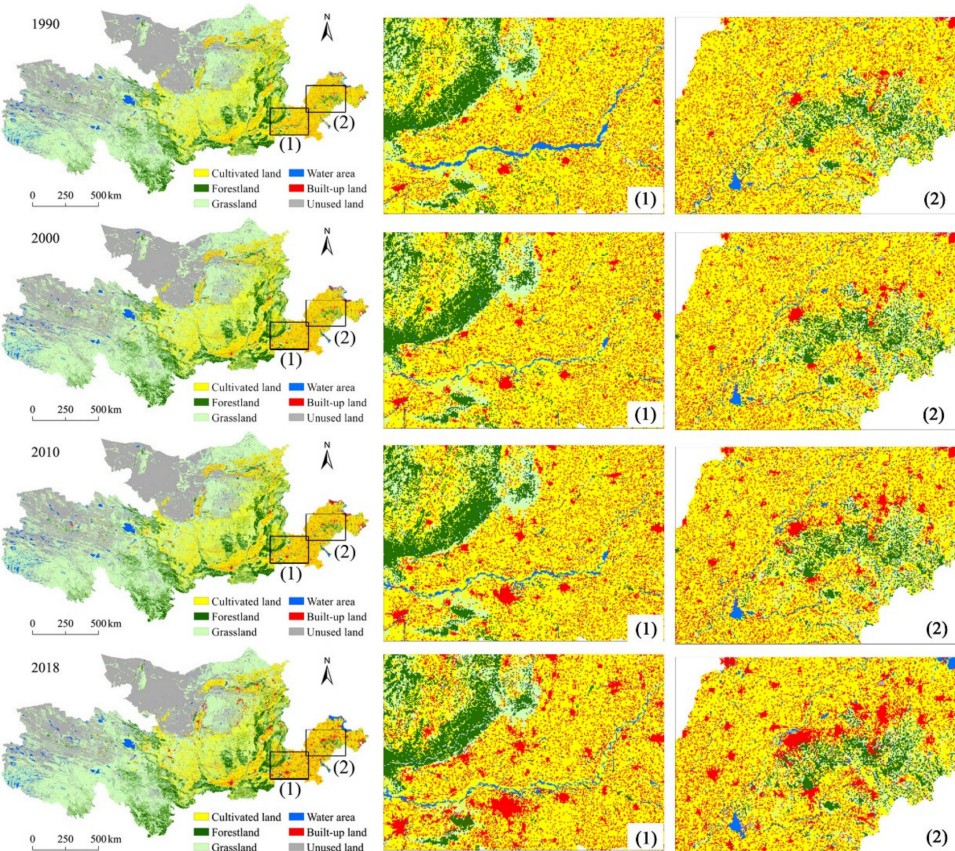

**Figure 3.** Land use distribution in the YRB from 1990 to 2018.

Table 2 shows that the proportions of various land use areas in the YRB have slightly changed in the past 30 years (1990–2018), but the varies in various land use areas are more significant. In the past 30 years, grassland accounted for the largest proportion of the land use structure within the research area (on average 41%). Unused land area accounted for ~27%; cultivated land accounted for ~18%; forestland accounted for ~9.7%; and the water and built-up land areas accounted for the smallest proportions, with ~2.5% and 3%, respectively. Overall, the cultivated land area sharply decreased by 15,289 km$^2$ at a rate of 4.2%. The unused land area has also decreased over the past 30 years. It decreased by 22,887 km$^2$ at a rate of 4.1%. Grassland accounts for the largest proportion in the study area. The grassland area insignificantly changed from 1990 to 2018, showing a fluctuating growth. It increased from 834,735 to 836,222 km$^2$ at a rate of 0.2%. From 1990 to 2018, the forestland area increased by 3174 km$^2$ at a rate of 1.6%. The water area steadily increased. It significantly increased in 2018, with an overall increase of 14%. Dongying and Binzhou are affected by marine economy and tidal flat planning of aquaculture waters, and the water area of the two cities has increased significantly, which is an important reason for the increase of the overall water area of the study area. Based on the promotion of reform and the "opening-up", the built-up land area significantly and rapidly increased from 1990 to 2018 by 24,376 km$^2$ at a rate of 64%. It is mostly distributed in the middle and lower reaches areas.

**Table 2.** Area and proportion of various land use types in the YRB from 1990 to 2018.

| Land Use Type | | Cultivated Land | Forestland | Grassland | Water Area | Built-Up Land | Unused Land |
|---|---|---|---|---|---|---|---|
| In 1990, | area/km² | 36,608 | 194,162 | 834,735 | 44,936 | 37,875 | 554,633 |
| | scale,/% | 18.01% | 9.55% | 41.07% | 2.21% | 1.86% | 27.29% |
| In 2000, | area/km² | 36,811 | 193,750 | 829,667 | 45,048 | 40,963 | 555,628 |
| | scale,/% | 18.11% | 9.53% | 40.81% | 2.22% | 2.01% | 27.33% |
| In 2010, | area/km² | 36,126 | 196,515 | 828,374 | 46,389 | 45,905 | 554,814 |
| | scale,/% | 17.77% | 9.67% | 40.74% | 2.28% | 2.26% | 27.29% |
| In 2018, | area/km² | 35,089 | 197,336 | 836,222 | 51,314 | 62,251 | 531,746 |
| | scale,/% | 17.28% | 9.72% | 41.20% | 2.53% | 3.07% | 26.20% |
| Amount of area change from 1990 to 2018 | | −1529 | 3174 | 1487 | 6378 | 24,376 | −22,887 |
| Rate of area change from 1990 to 2018 | | −4.18% | 1.63% | 0.18% | 14.19% | 64.36% | −4.13% |

### 3.1.2. Change Characteristics of the ESV

Land use change affects the ESV [55]. From 1990 to 2018, the ESV of the YRB was USD 591 billion, USD 590 billion, USD 594 billion, and USD 606 billion, respectively. The ESV first declined and then rose, and the overall growth is slow (Table 3). The ESV of cultivated land changed significantly. Over the past 30 years, it has decreased by USD 3 billion at a change rate of ~4%. The change of the forestland ESV is the same as that of its area. It fluctuated and slowly increased from 1990 to 2018. Over the past 30 years, ESV has increased by USD 2 billion at a growth rate of ~2%. Grassland contributed the most to the ESV of the YRB, with USD 246 billion, USD 245 billion, USD 245 billion, and USD 247 billion in the four periods, respectively. The results show that the overall change in the grassland ESV from 1990 to 2018 is small. The value is the same as that of forestland, showing a fluctuating increase. The ESV increased by USD 400 million at a rate of ~1%. After 2000, the forestland and grassland areas slightly increased and the ESV of the water area significantly raised. In addition, ESV has increased by USD 16 billion in the past 30 years at a rate of ~14%. Unused land contributed the least to the ESV in the study area. Due to the demand for rapid urban development, the unused land area significantly decreased, and its ESV also decreased. In the last 30 years, the ESV has decreased by USD 800 million at a rate of ~4%. Generally speaking, the areas of the land use types that strongly contribute to the ESV, i.e., forestland, grassland, and water area, show a fluctuating increase, whereas the built-up land area, which does not contribute to the increase in the ESV, increased rapidly, resulting in the decrease in the unused land area with a small contribution to the ESV, due to occupation of cultivated land. This also led to a decrease in the ESV of cultivated land, which greatly contributes to the overall ESV of the YRB. However, the increase in the ESV of grassland, forestland, and the water area compensates for the decline in the ESV of cultivated land. Therefore, over the past 30 years, the overall ESV in the YRB has fluctuated. It has increased by ~USD 15 billion at a rate of ~3%.

The average ESV of each prefecture-level city in the YRB was further visualized using ArcGIS 10.5 (ESRI Inc., Redlands, CA, USA), and a spatiotemporal map of the distribution of the ESV was obtained (Figure 4). The ESVs of all prefecture-level cities in the YRB show a spatial pattern of high middle reaches and low middle reaches in the upper and lower reaches. From 1990 to 2000, the average ESV of Xinxiang, Xianyang, Hebi and Pingliang decreased by one interval value, whereas the ESV of Haidong, Binzhou, Dongying increased by one interval value and that of Jining increased by three interval values. From 2000 to 2010, the average ESV of Hainan, Haibei, Gannan and Binzhou decreased by ~USD 600, whereas the ESV of Xianyang increased by one interval value. The average ESV of other prefecture-level cities insignificantly changed. From 2010 to 2018, the ESVs of Hainan, Haibei, Gannan, Jining, Weifang, Qingdao, Binzhou and Dongying increased, whereas those of Xianyang, Yuncheng, Luliang, Taiyuan, Zhengzhou, and Jiaozuo decreased. From

1990 to 2018, notable changes in the ESVs of prefecture-level cities in the YRB mainly occurred in the upper and lower reaches areas.

**Table 3.** Changes in the ESVs of various land use types in the YRB from 1990 to 2018 (USD $10^8$).

| Land Use Type | | Cultivated Land | Forestland | Grassland | Water Area | Unused Land | Sum |
|---|---|---|---|---|---|---|---|
| 1990 | ESV | 731.72 | 1381.34 | 2464.56 | 1138.24 | 195.05 | 5910.91 |
| | Scale/% | 12.38% | 23.36% | 41.70% | 19.26% | 3.30% | 100.00 |
| 2000 | ESV | 735.88 | 1378.41 | 2449.60 | 1141.08 | 195.40 | 5900.36 |
| | scale/% | 12.47% | 23.36% | 41.52% | 19.34% | 3.31% | 100.00 |
| 2010 | ESV | 722.06 | 1398.08 | 2445.78 | 1175.05 | 195.11 | 5936.08 |
| | scale/% | 12.16% | 23.55% | 41.20% | 19.80% | 3.29% | 100.00 |
| 2018 | ESV | 701.16 | 1403.92 | 2468.95 | 1299.80 | 187.00 | 6060.83 |
| | scale/% | 11.57% | 23.16% | 40.74% | 21.45% | 3.08% | 100.00 |
| Amount of ESV change from 1990 to 2018 | | −30.56 | 22.58 | 4.39 | 161.56 | −8.05 | 149.92 |
| Change rate of ESV from 1990 to 2018 | | −4.18% | 1.63% | 0.18% | 14.19% | −4.13% | 2.54% |

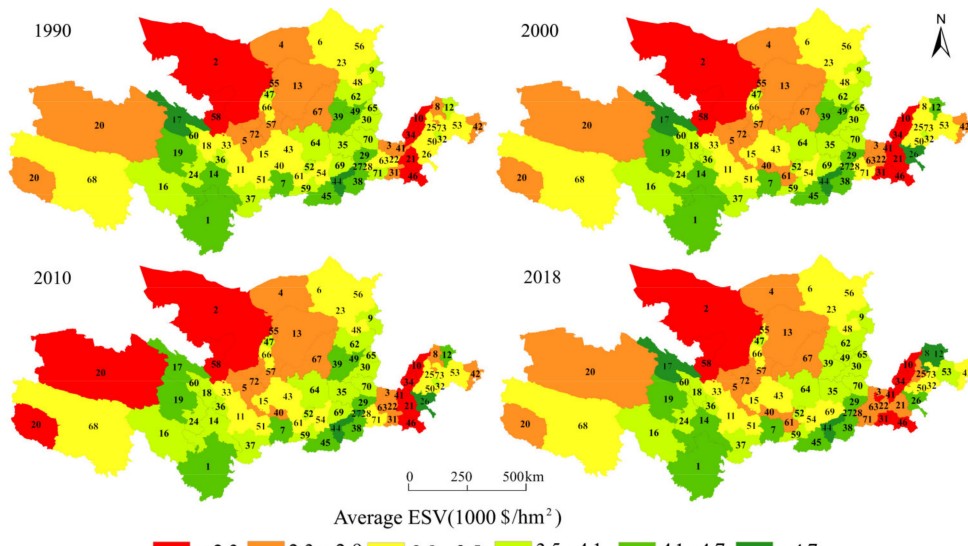

Annotation: 1. Aba prefecture 2. Alxa league 3. Anyang 4. Bayannur 5. Baiyin 6. Baotou 7. Baoji 8. Binzhou 9. Datong 10. Dezhou 11. Dingxi 12. Dongying 13. Ordos 14. Gannan 15. Guyuan 16. Golog 17. Haibei 18. Haidong 19. Hainan 20. Hainan 21. Heze 22. Hebi 23. Huhehaote 24. Huagnan 25. Jinan 26. Jining 27. Jiyuan 28. Jiaozuo 29. Jincheng 30. Jinzhong 31. Kaifeng 32. Laiwu 33. Lanzhou 34. Liaocheng 35. Linfen 36. Linxia 37. Longnna 38. Luoyang 39. Luliang 40. Pingliang 41. Puyang 42. Qingdao 43. Qingyang 44. Sanmenxia 45. Sahngluo 46. Shangqiu 47. Shizuishan 48. Shuozhou 49. Taiyuan 50. Taian 51. Tianshui 52. Tongchuan 53. Weifang 54. Weinan 55. Wuhai 56. Wulanchabu 57. Wuzhong 58. Wuwei 59. Xian 60. Xining 61. Xianyang 62. Xinzhou 63. Xinxiang 64. Yanan 65. Yangquan 66. Yinchuan 67. Yulin 68. Yushu 69. Yuncheng 70. Changzhi 71. Zhengzhou 72. Zhongwei 73. Zibo

**Figure 4.** Spatiotemporal distribution of the ESV in the prefecture-level cities of the YRB from 1990 to 2018.

## 3.2. Sensitivity Analysis

The coefficient of sensitivity (CS) can be used to determine the extent to which the ESV varied depending on the ecosystem value coefficient [56]. The sensitivity index of the ESV in the YRB from 1990 to 2018 was calculated by adjusting the ecosystem value coefficients of different land use types by ±50% (see Table 4). Based on the analysis in Table 4, the sensitivity index of the five land use types is smaller than 1; thus, the research results are reliable. The sensitivity index of unused land is smaller than 0.1, indicating that its accuracy of the VC insignificantly affects the total ESV in the study area. The grassland sensitivity index ranges from 0.407–0.417, indicating that for every 1% increase in the VC of

grassland, the ESV in the study area will increase by 0.407–0.417%, far exceeding the other four types. The CS values of forestland, the water area, and cultivated land are greater than 1%, indicating that these land use types significantly affect the ESV in the regional study. The results show that the sensitivity of ecosystem value coefficient in regional research is grassland > forestland > water area > cultivated land > unused land.

**Table 4.** Sensitivity index values of various ESs in the YRB.

| Land Use Type | 1990 | 2000 | 2010 | 2018 |
|---|---|---|---|---|
| Cultivated land | 0.124 | 0.125 | 0.122 | 0.116 |
| Forestland | 0.234 | 0.234 | 0.236 | 0.232 |
| Grassland | 0.417 | 0.415 | 0.412 | 0.407 |
| Water area | 0.193 | 0.193 | 0.198 | 0.214 |
| Unused land | 0.033 | 0.033 | 0.033 | 0.031 |

### 3.3. Changes in Land Use and ESV under Multiple Scenarios

#### 3.3.1. Accuracy Verification

To determine the credibility of the model in predicting the future land use pattern, we compared the simulated 2018 land use map and actual land use types in the YRB in 2018. The calculated kappa value was 0.89. The error of the cultivated land and built-up land areas was ~0.04%, the error of the forestland area was ~0.13%, that of the grassland area was ~0.27%, and the error of the unused land area was ~0.5%. Therefore, the errors of all land use type areas were ≤1%. Thus, the model passes the accuracy test and has a high degree of reliability.

#### 3.3.2. Multi-Scenario Simulations

According to the actual situation of the YRB, three land use change scenarios—natural development, cultivated land protection and ecological protection—in 2026 were simulated (see Figure 5). The map of the land use distribution according to the three scenarios shows that the distribution of different categories in the YRB insignificantly differs. In the natural development and ecological protection scenarios, the cultivated land, forestland, grassland, along with water areas are the same. In the ecological protection scenario, the built-up land area is 4064 km$^2$ smaller than that in the natural development scenario, the unused land area is 4064 km$^2$ larger. In the cultivated land protection scenario, the cultivated land area is greater than that in the ecological protection and natural development scenarios. The grassland, forestland, and water areas are the same. In the cultivated land protection scenario, the built-up land area is 10,000 km$^2$ smaller than that in the other two scenarios, but the increase in the cultivated land area is more notable.

Table 5 shows the further analysis of various ESVs under the three scenarios. Compared with the ESVs of various land use types in 2018, among the three scenarios, the change trend of ESV in the natural development and the ecological protection scenario is the same: the ESV of the water area significantly decreases, the ESVs of cultivated land, forestland, grassland, and unused land slightly increase; the cultivated land protection scenario of 2026, the ESV of water area and forestland decreases, while that of cultivated land, grassland and unused land increases. Compared with the natural development and ecological protection scenarios, the ESV of cultivated land and unused land was higher than that of the natural development and ecological protection scenarios in the cultivated land protection scenario. Compared with 2018, the total ESV of the YRB slowly increases in natural development and ecological protection scenarios, whereas it decreases in the cultivated land protection scenario, where the ESV value is ~USD 200 million smaller. Overall, the total ESV of each scenario varies more obvious. The ESV of the ecological protection scenario is higher than other two scenarios, specifically, ~USD 150 million and ~USD 800 million higher than in the natural development and cultivated land protection scenario, respectively.

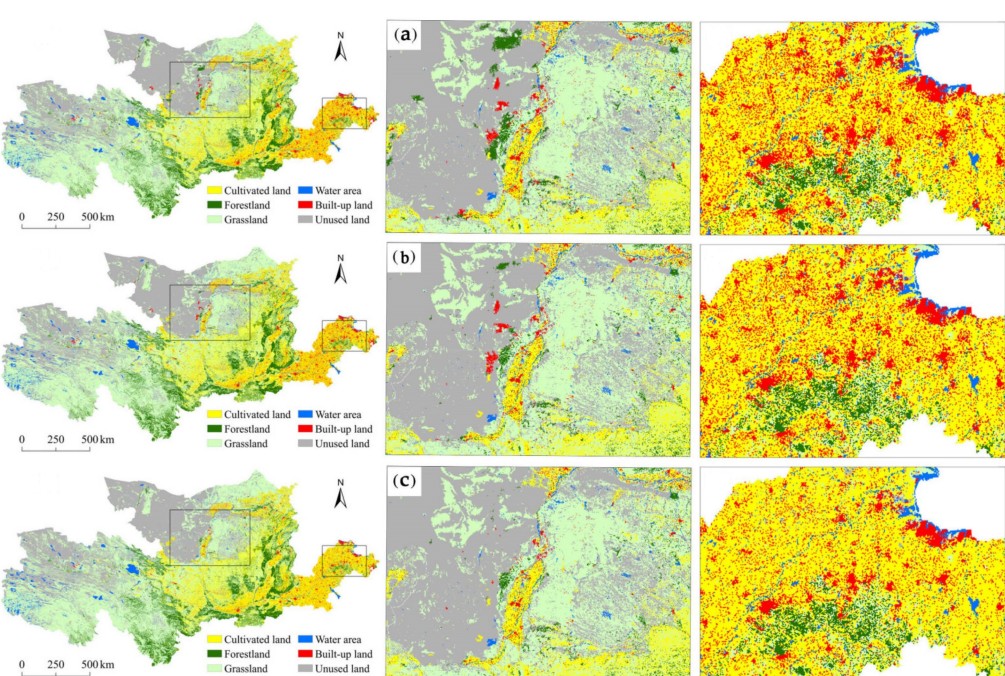

**Figure 5.** Simulation of the land use in the YRB in 2026 based on three scenarios. (**a**) 2026 natural development; (**b**) 2026 ecological protection; (**c**) cultivated land protection.

**Table 5.** Simulation of the ESVs of various land use types in 2026 under three scenarios (USD $10^8$).

| Land Use Type | | Cultivated Land | Forestland | Grassland | Water Area | Unused Land | Amount To |
|---|---|---|---|---|---|---|---|
| In 2018 | ESV | 701.16 | 1403.92 | 2468.95 | 1299.80 | 187.00 | 6060.83 |
| | scale/% | 11.57% | 23.16% | 40.74% | 21.45% | 3.09% | 100.00% |
| Natural development | ESV | 710.68 | 1418.46 | 2535.21 | 1207.50 | 193.04 | 6064.88 |
| | scale/% | 11.72% | 23.39% | 41.80% | 19.91% | 3.18% | 100.00% |
| Ecological protection | ESV | 710.68 | 1418.46 | 2535.21 | 1207.50 | 194.46 | 6066.31 |
| | scale/% | 11.72% | 23.38% | 41.79% | 19.90% | 3.21% | 100.00% |
| Cultivated land protection | ESV | 736.62 | 1383.88 | 2535.21 | 1207.50 | 195.56 | 6058.77 |
| | scale/% | 12.16% | 22.84% | 41.84% | 19.93% | 3.23% | 100.00% |

The natural development scenario continues the land demand of the YRB against the background of urbanization, and the ESV slightly increases. The coordination of ecological protection scenarios limits the transformation and reduction of ecological land, and better meets the requirements of YRB during urbanization and socioeconomic enhancement. The cultivated land protection scenario highlights the correlation between cultivated land protection and urbanization, focuses on alleviating the contradiction between urbanization and cultivated land occupation, and ensures that the cultivated land area is not reduced while meeting the demand for built-up land during rapid urbanization.

## 4. Discussion

The results show that the area of the YRB significantly changed during the study period, and the built-up land area rapidly expanded. However, with the implementation of policies and measures related to ecological protection and the enhancement of people's environmental protection concepts, areas of ecological land, such as grassland, forestland, and water area, have generally increased. Therefore, the ESV of ecological land also increases and the total ESV of the YRB increases slowly. This shows that increasing ecological land such as forestland, grassland and water area can effectively improve regional ESV [57]. To optimize the land use composition to raise the watershed ESV, the different land use patterns in the basin were simulated in this study based on various scenarios. The 2026

ESV under the cultivated land protection scenario was lower than that in 2018, and the ESVs of both the natural development and ecological protection scenario were higher than those in 2018. The highest value was obtained for the ecological protection scenario. The results are beneficial for providing guidance for the future development and construction of the basin from the perspective of optimizing ESV.

Land use change is considered to be one of the most important factors affecting global environmental change [58–60]. With rapid population growth, urbanization and climate change, the services of global ecosystems are continuously degraded, which further affects ecosystem services to humans [61,62]. Therefore, quantitative assessment of the impact of land use change on ecosystem service value is of great significance in guiding the sustainable development of the global ecological environment [63,64]. We analyzed the river basin with respect to land use changes over the past 30 years. The results reveal the spatiotemporal changes of the ESV. The variety of land use in the basin is consistent with the change in the ESV, showing a fluctuating increase. The ESV of the watershed decreased during 1990–2000 and slowly increased after 2000. This concurs with the research results of Deng et al. [65], who reported that the ESV in the YRB was increasing. These changes can be attributed to the implementation of the large-scale "return cultivated land to forest to grass project" in 1999 [66,67], greatly alleviating the declining trends in the upper and middle reaches of the YRB. Ecosystem service level is gradually being restored and the ESV gradually increasing, which shows that the increased restoration of forestland, grassland and other ecological land can significantly improve the level of ecosystem services. This conclusion is in accordance with the findings of Liu et al. [68]. The results also show that forestland, grassland, and other ecological land use types are the main contributors to the ESV in the YRB, mainly because the areas of forestland and grassland in the basin are large and the increase in the ESV of forestland and grassland determines the increase in the overall ESV in the YRB. To verify the reliability of the above-mentioned results, we compared them with the results of a recent ESV study of Wu et al. [69]. Additionally, the increase in the ecosystem areas of forestland and grassland in the two watersheds is the main reason for the increase that ESV in the Yellow River and Yangtze River basins. Although the ESV of YRB has generally increased, the region still faces ecological threats. From the point of view of the land use distribution in the basin (Figure 3), the lower reaches are dominated by cultivated land. Due to urbanization, a large area of cultivated land is occupied, and the ESV of cultivated land decreases. This shows that the reduction in cultivated land changes the ESV in the lower reaches [70]. Therefore, the land use structure in the watershed must be rationally adjusted under the guidance of ecological conservation and high-quality development of the YRB, such as with respect to soil erosion and ecological destruction.

When considering different development goals, regional land use patterns will be different. It is of great significance to explore the land use structure under various development goals for regional sustainable development and efficient utilization of resources. According to the different service values provided by different ES types in the three scenarios (Figure 6), the ESV provided by forestland, grassland and water area was higher in the three scenarios. It was further found that forestland, grassland and water area played an important role in maintaining biodiversity diversity, hydrological regulation, ecosystem maintenance, etc. The natural environment of the upper, middle, and lower reaches of the YRB is complicated. The upper reaches of the Yellow River are the main source of water supply, so it is particularly important to strengthen the water conservation function of the upper reaches. Therefore, attention should be paid to ecological land with a strong supporting service capacity of grassland, forestland and water area in the upper reaches, and vegetation restoration and ecological protection restoration projects should be accelerated. Soil erosion is serious in the middle reaches of the YRB, so they are particularly important with respect to soil retention service. Grassland and forestland provide the strongest supporting service capacity, indicating that the middle reaches should also pay attention to ecological land use. We will move faster to protect forestland and grassland

and enhance our ability to conserve water and soil. The lower reaches of the YRB are densely populated and are an important grain producing area in China. The development of urbanization has occupied a large amount of cultivated land, food security is threatened, and it is necessary to slow down the rate of decline of cultivated land area. The cultivated land protection scenario is suitable for the development of the lower reaches. Although the ESV has declined, it can promote the coordinated development of ecological protection and population economy in the lower reaches. In general, the three development scenarios have reference significance for the future ecological construction and built-up land expansion of the basin. The simulation results were combined with the actual situation of different regions to provide guidance for different regions to adjust the future land use structure, thus accelerating the realization of watershed ecological protection and high-quality development of the national strategic goals.

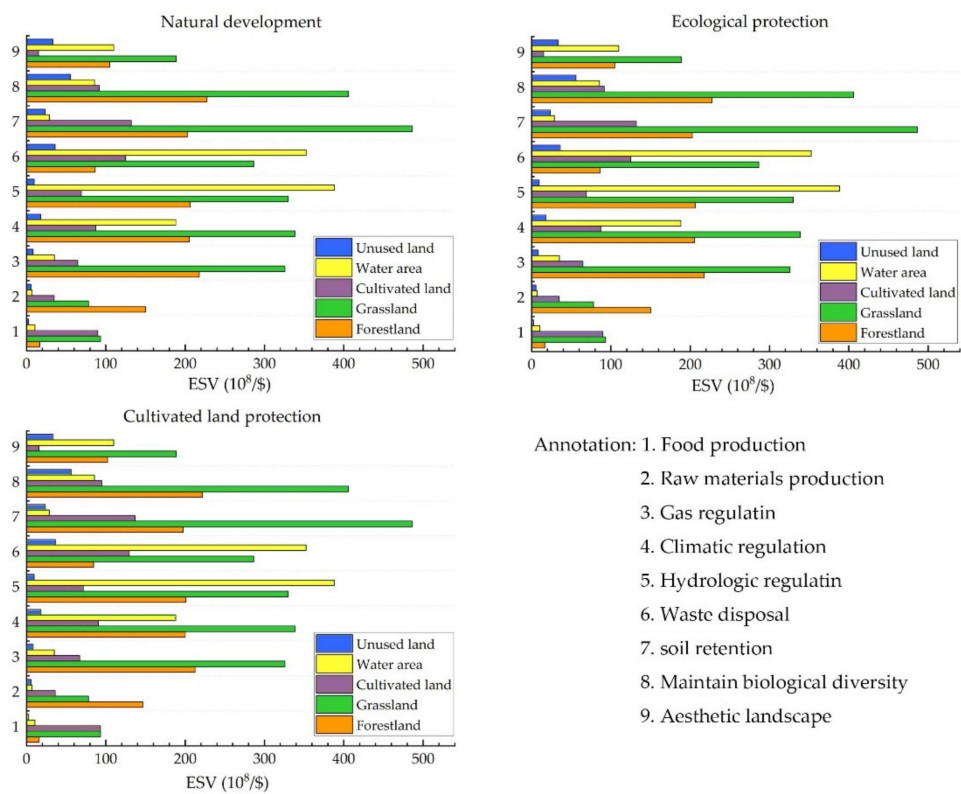

**Figure 6.** ESV of different land use types under different scenarios.

In addition, the PLUS model was used in this study, which has a higher simulation prediction accuracy for larger areas and can better predict the evolution of future land use patterns in the YRB. Because this model is more flexible in handling multiple types of land use pattern change mechanisms than traditional prediction models, many researchers have made forecasts of land use change at different scales utilizing this model. For example, Shi et al. [71] used the PLUS model to depict the actual land use change of other river basins more accurately, and to analyze future land use patterns of river basins under different scenarios. Based on this model, Zhao et al. [72] predicted the land use structure in Wuhan under different scenarios in 2035 and compared their results with those of other models, achieving a higher simulation accuracy and more similar landscape patterns, providing guidance for urban sustainable development. In summary, researchers have obtained more accurate and reliable simulation results under various scenarios using the PLUS model, providing substantial guidance for decision makers to manage land use patterns in different development goals and confirming scientific nature of the model. This research also used this model to imitate the land use patterns under three scenarios in the YRB in 2026, which confirms the reliability of this study.

Land use change is a complex dynamic process, and an analysis of the present situation of land use compared to the past cannot fully express the qualitative change of land use types [73], and there is a lack of consideration for the impact of multiple development goals. By analyzing the past and simulating future land use patterns under different scenarios, this study discusses the impact of future land use structure change on ESV under the influence of different development goals. The results enrich the research on the impact of land use change in the YRB on future ESV. The results can also provide a reference for the study of land use change and ESV in other large river basins. However, this study has a limitation, which must be addressed in the future. Due to the complexity of the social and natural composition of built-up land, the ESV of built-up land is ignored during the calculation of the ESVs among various types of land use, which affects the total ESV of the basin. The estimation of the ESV of built-up land needs to be further studied.

**5. Conclusions**

In the context of global ecological sustainable development needs, quantifying the impact of past and future land use changes on the ESV can help to better understand the relationship between land use and ecosystem services. We reasonably describe the evolution of three different land use patterns in the basin in the future through scenario analysis. Comparing the ESV under different scenarios, it is found that the ESV under the ecological protection scenario is the highest, and urbanization is not too limited compared with the cultivated land protection scenario. Therefore, we believe that the ecological protection scenario may be more suitable for the future development of the basin from the perspective of improving the ESV and the YRB's adherence to the ecological priority development concept. However, the natural conditions in the upper, middle, and lower reaches of the YRB vary greatly, and the priorities for ecological protection and social construction and development are different. From the perspective of adjusting measures to local conditions and implementing classified policies, the three scenarios will have reference values for the future adjustment of regional land use structure and optimization of ecosystem services in the YRB. To better understand the correlation between land use and ESV and to maximize watershed ESV, we will make further efforts in future research to integrate land use and ecosystem data, and study the spatial response of watershed ESV by establishing a grid, so as to further improve the analytical accuracy of ESV measurement.

**Author Contributions:** Project administration and funding acquisition, P.Z.; writing—review and editing, P.Z. and W.J.; writing—original draft and methodology, Y.L.; investigation, D.Y., M.S., Y.H. and Y.Z. All authors have read and agreed to the published version of the manuscript.

**Funding:** This research has been supported by the National Natural Science Foundation of China, grant number 41601175, 41801362, the 2018 Young Backbone Teachers Foundation from Henan Province, grant number 2020GGJS114, the Program for Innovative Research Talent in University of Henan Province, grant number 20HASTIT017, 2021 Project of Henan Soft Science Funds, grant number 212400410250, the Youth Natural Science Foundation of Henan Province, grant number 202300410077, 2020 Philosophy and Social Science Planning Project of Henan Province, grant number 2020BJJ020.

**Data Availability Statement:** Not applicable.

**Acknowledgments:** We also thank the Geographical Science Data Center of The Greater Bay Area for providing the relevant data in this study.

**Conflicts of Interest:** The authors declare no conflict of interest.

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
