# Peer review of "Multi-Scenario Simulation of Land Use Changes with Ecosystem Service Value in the Yellow River Basin"

_land, doi:10.3390/land11070992_

Round 1

Reviewer 1 Report

Through a round of revisions, I can see the readability of the manuscript has improved a lot. I appreciate the authors' energy and effort on it. I think it will be a good contribution to our Journal and I recommend to accept the paper.

Author Response

Thanks for your suggestions, which is very helpful to the improvement of our article.

Reviewer 2 Report

The manuscript "Multi-scenario Simulation of Land Use Changes with Ecosystem Service Value in the Yellow River Basin" evaluates the land-use changes occurring in the Yellow River Basin in 1990, 2000, 2010, and 2018 and their corresponding implications on the ecosystem service value. Moreover, the study simulates future land-use alterations under three different scenarios with the PLUS model and their intrinsic repercussions on the regional ecosystem service value in an attempt to optimize the land use structure for 2026. The cultivated land assumes only wheat, corn, and rice as main crops sown in the Yellow River Basin.

The authors argue that, while similar studies exist for various large-scale areas, there is a lack of assessments and multi-scenario simulations of complex land-use patterns and alterations of river basins, in general. In this context, the novelty of the manuscript stems from evaluating the former and ongoing land-use alterations in the Yellow River Basin, especially important due to its characteristics as an ecological barrier and major grain producing zone in China.

Generally, the manuscript is descriptive, concise, and well-written. Reproducibility has been granted by thoroughly describing the steps undertaken in the study. The manuscript concludes by stating its limitations, outlook, recommendations, and need for future work.

However, punctuation errors and a need for rephrasing still exist. As the reader progresses through the manuscript, the grammar worsens, risking readability. Please consider an English editor, there are plenty of paragraphs which need rephrasing, especially in the Results and Discussion chapters.

Abstract

L32: The sentence “The 2026 land use pattern of the YRB was simulated under three different scenarios using the PLUS model.” is redundant, as it has been elaborated before, in lines 27-30. Please consider erasing it.

Introduction

L46: “ESV as is a tool that used to appraise (…)”.

L64, 67: The punctuation period should be used after “et al.”, not after the reference.

Materials and Methods

L138-139: “According to the study on by Xie et al. (…)”.

L143-144: Both the numerator and denominator of eq. (1) are annotated Mi in the text.

L148: The punctuation period should be used after “et al.”, not after the reference.

L150: I believe Ai refers to the area of the class i land use type and VCj to the ESV per unit area of the class j land use type. Would be less confusing if Ai would be written Aj?

L154: It is not clear how the market price of 0.36$/kg was reached for three different crops under almost three decades. Please elaborate.

L194: Please consider rephrasing: “The lower reaches of YRB is are an important grain producing area in my country China.”

Results

L212: “(…) pattern is an a vital source of (…)”

L317: The sentence “Simulate the natural development, cultivated land protection, and ecological protection scenarios of YRB in 2026 (see Figure 5).” needs rephrasing.

L340-341: “(…) whereas the it decreases (…) that ESV value is (…)”. Please revise.

L342: “The total ESV of each scenario insignificantly differs.” Could you explain in comparison to what? Differences of 150, 200, and 800 Mill seem quite significant in the right context.

Discussion

Figure 6. There is annotation called “Keep the soil”, could you elaborate or change? Maybe you refer to soil quality, but then please explain in which terms. Additionally, why is the unit of ESV here 108/$?

L454-461: Seven lines and a half for one, very long sentence, which intertwines so much that it is not clear any more what the subject is. Please rephrase.

Conclusion

L486: Could you please elaborate, what would a microscopic perspective be in the context of the Yellow River Basin?

Author Response

Point–by–point responses

Reviewer #2

The manuscript "Multi-scenario Simulation of Land Use Changes with Ecosystem Service Value in the Yellow River Basin" evaluates the land-use changes occurring in the Yellow River Basin in 1990, 2000, 2010, and 2018 and their corresponding implications on the ecosystem service value. Moreover, the study simulates future land-use alterations under three different scenarios with the PLUS model and their intrinsic repercussions on the regional ecosystem service value in an attempt to optimize the land use structure for 2026. The cultivated land assumes only wheat, corn, and rice as main crops sown in the Yellow River Basin.

The authors argue that, while similar studies exist for various large-scale areas, there is a lack of assessments and multi-scenario simulations of complex land-use patterns and alterations of river basins, in general. In this context, the novelty of the manuscript stems from evaluating the former and ongoing land-use alterations in the Yellow River Basin, especially important due to its characteristics as an ecological barrier and major grain producing zone in China.

Generally, the manuscript is descriptive, concise, and well-written. Reproducibility has been granted by thoroughly describing the steps undertaken in the study. The manuscript concludes by stating its limitations, outlook, recommendations, and need for future work.

However, punctuation errors and a need for rephrasing still exist. As the reader progresses through the manuscript, the grammar worsens, risking readability. Please consider an English editor, there are plenty of paragraphs which need rephrasing, especially in the Results and Discussion chapters.

Response: Thank you for your suggestion, which is very helpful to improve the quality of the article. We have also carefully revised it in strict accordance with your reviews. At the same time, we invited a professional English teacher in the Editage (https://app.editage.cn) to revise the grammar of our article.

Specific comments

Comment 1:L32: The sentence “The 2026 land use pattern of the YRB was simulated under three different scenarios using the PLUS model.” is redundant, as it has been elaborated before, in lines 27-30. Please consider erasing it.

Response: Thank you for your comments. We have removed this sentence as you suggested (Corresponding to line 32 in the revised manuscript). The details are as follows.

Line32: The ESV of the YRB has slowly increased by ~$15 billion over the past 30 years. The ESV obtained under the ecological protection scenario is the highest.”

Comment 2:L46: “ESV as is a tool that used to appraise (…)”.

Response: Thank you for your comments. We have made the modification according to your suggestion (Corresponding to line 45 in the revised manuscript). The details are as follows.

Line45: “ESV is a tool that used to appraise the strength of ES”

Comment 3:L64, 67: The punctuation period should be used after “et al.”, not after the reference.

Response: Thank you for your comments. We have corrected these errors (Corresponding to lines 62-66、138、147、383、389、395、441、443 in the revised manuscript). The details are as follows.

Line62-66: “Costanza et al. [22] established the global ESV equivalent factor table in 1997, laying a theoretical foundation for relevant research. Subsequently, researchers have studied the changes in the ESV at different scales (global [23], national [24], and coastal [25]). According to Costanza et al. [22], Xie et al. [26] formulated an equivalent table of the ESV per unit area for China's terrestrial ecosystem.”

Line138: “According to the study on by Xie et al. [50],”

Line147: “With reference to the equivalent value scale propounded by Xie et al. [50], ”

Line383: “The concurrent with the research results of Deng et al. [65],”

Line389: “This conclusion is accordant to the findings of Liu et al. [68].”

Line395: “we compared them with the results of a recent ESV study of Wu et al. [69].”

Line441:”For example, Shi et al. [71] used the PLUS model to depict the actual land use change of other river basin more accurately,”

Line443:“Based on this model, Zhao et al. [72] predicted the land use structure in Wuhan under different scenarios in 2035 and compare with other models,”

Comment 4:L138-139: “According to the study on by Xie et al. (…)”.

Response: Thank you for your comments. We have corrected these errors (Corresponding to lines138、147 in the revised manuscript). The details are as follows.

Line138: “According to the study on by Xie et al. [50],”

Line147: “With reference to the equivalent value scale propounded by Xie et al. [50], ”

Comment 5:L143-144: Both the numerator and denominator of eq. (1) are annotated Mi in the text.

Response: Thank you for your comments. We have modified this (Corresponding to lines142-143 in the revised manuscript). The details are as follows.

Line142-143: “mi represents the economic output value of the ith grain crop, and Mi represents the total sown area of the ith grain crop. ”

Comment 6:L148: The punctuation period should be used after “et al.”, not after the reference.

Response: Thank you for your comments. We have corrected this error (Corresponding to line147 in the revised manuscript). The details are as follows.

Line147: “With reference to the equivalent value scale propounded by Xie et al. [50], ”

Comment 7:L150: I believe Ai refers to the area of the class i land use type and VCj to the ESV per unit area of the class j land use type. Would be less confusing if Ai would be written Aj?

Response: Thank you for your comments. We have changed Ai to Aj according to your suggestion (Corresponding to line142-143 in the revised manuscript). The details are as follows.

Line142-143: “where Aj mean the area of the class j land use type and VCj express the ESV per unit area of the class j land use type.”

Comment 8:L154: It is not clear how the market price of 0.36$/kg was reached for three different crops under almost three decades. Please elaborate.

Response: Thank you for your comments. We calculated the average prices of wheat, corn and rice by referring to China's Agricultural Product Price Survey Yearbook and the statistical yearbooks of provinces in the Yellow River Basin, and found that the market price of wheat, corn and rice per unit area in the study area is 2.36 yuan/kg. According to the data, the national average exchange rate of RMB against USD in 2018 is 0.15124. Finally, the market price of the three crops was calculated to be 0.35692$/kg, with two decimal digits remaining 0.36$/kg.

Comment 9:L194: Please consider rephrasing: “The lower reaches of YRB is are an important grain producing area in my country China.”

Response: Thank you for your comments. We have revised this sentence to "The lower reaches of YRB is an important grain producing area in China." (Corresponding to lines193-194 in the revised manuscript). The details are as follows.

Line193-194: “The lower reaches of YRB is an important grain producing area in China.”

Comment 10:L212: “(…) pattern is an a vital source of (…)”

Response: Thank you for your comments. We have changed an to a (corresponding to line 211 of the revised document). The details are as follows:

Line211: “The evolution of the land use pattern is a vital source of information for understanding ecological change and the impacts of human activities”

Comment 11:L317: The sentence “Simulate the natural development, cultivated land protection, and ecological protection scenarios of YRB in 2026 (see Figure 5).” needs rephrasing.

Response: Thank you for your comments. We have rephrased this sentence. Change to "According to the actual situation of the YRB, three land use change scenarios of natural development, cultivated land protection and ecological protection in 2026 were simulated (see Figure 5)." (corresponding to lines 316-318 of the revised document). The details are as follows:

Line316-318: “According to the actual situation of the YRB, three land use change scenarios of natural development, cultivated land protection and ecological protection in 2026 were simulated. (see Figure 5).”

Comment 12:L340-341: “(…) whereas the it decreases (…) that ESV value is (…)”. Please revise.

Response: Thank you for your comments. We have rephrased this sentence. Change to "Compared with 2018, the total ESV of the YRB slowly increases in natural development and ecological protection scenarios, whereas the it decreases in the cultivated land protection scenario, that ESV value is ~$200 million smaller." (corresponding to lines 338-341 of the revised document). The details are as follows:

Line338-341: “Compared with 2018, the total ESV of the YRB slowly increases in natural development and ecological protection scenarios, whereas the it decreases in the cultivated land protection scenario, that ESV value is ~$200 million smaller.”

Comment 13:L342: “The total ESV of each scenario insignificantly differs.” Could you explain in comparison to what? Differences of 150, 200, and 800 Mill seem quite significant in the right context.

Response: Thank you for your comments. “The total ESV of each scenario insignificantly differs.” indicates that there is little difference between natural development, cultivated land protection and ecological protection scenarios in Table 5 and ESV in 2018. This is because we believe that the difference of us $150 million, US $200 million and US $800 million is not very big considering that the ESV of the Yellow River Basin increased by about US $15 billion over the past three decades from 1990 to 2018. However, these differences are important to highlight the advantages of each situation, so we restated them. " Overall, the total ESV of each scenario varies more obvious. The ESV of the ecological protection scenario is higher than other two scenarios, specifically, ~$150 million and ~$800 million higher than natural development and cultivated land protection scenario, respectively.”(corresponding to lines 338-341 of the revised document). The details are as follows:

Line341-344: “Overall, the total ESV of each scenario varies more obvious. The ESV of the ecological protection scenario is higher than other two scenarios, specifically, ~$150 million and ~$800 million higher than natural development and cultivated land protection scenario, respectively.”

Comment 14: Figure 6. There is annotation called “Keep the soil”, could you elaborate or change? Maybe you refer to soil quality, but then please explain in which terms. Additionally, why is the unit of ESV here 108/$?

Response: Thank you for your comments. We refer to the classification table of China ecosystem service types in Xie et al. The soil retention contrast here is the Costanza classification "erosion control can maintain sediment, soil formation, nutrient cycling", which is defined as accumulation of organic matter and the role of vegetation root matter and organisms in soil conservation, nutrient cycling and accumulation. In addition, it was changed to "soil Retention" by referring to previous studies [1]. In addition, Figure 6 calculates ESV values of nine ecosystem services, which are consistent with the UNITS of ESV values in Table 3 and Table 5 of the paper. We modify Figure 6 (corresponding to modified file Table 1, Line 420 and Figure 6). The details are as follows:

Table 1. ESV coefficients for the YRB.

Ecosystem Service Value ($ hm-2a-1)

ecosystem-type

Forest

land

Grass

land

Cultivated land

Water

area

Unused land

Provisioning

Services

Food

production

83.49

108.79

253.00

225.17

5.06

Raw materials

production

753.94

91.08

98.67

149.27

10.12

Subtotal

837.43

199.87

351.67

374.44

15.18

Regulating

Services

Gas regulatin

1092.96

379.50

182.16

738.76

15.18

Climatic regulation

1029.71

394.68

245.41

3949.33

32.89

Hydrologic regulatin

1034.77

384.56

194.81

8149.12

17.71

Waste disposal

435.16

333.96

351.67

7400.24

65.78

Subtotal

3592.60

1492.70

974.05

20237.45

131.56

Supporting

Services

Soil retention

1017.06

566.72

371.91

607.20

43.01

Maintain biological diversity

1141.03

473.11

258.06

1801.36

101.20

Subtotal

2158.09

1039.83

629.97

2408.56

144.21

Cultural Services

Aesthetic landscape

526.24

220.11

43.01

2309.89

60.72

Subtotal

526.24

220.11

43.01

2309.89

60.72

Amount to

7114.35

2952.51

1998.70

25330.33

351.67

Line420: “so it is particularly important for soil retention service.”

[1] Xiao, R.; Lin, M.; Fei, X.F.; Li, Y.S.; Zhang, Z.H.; Meng, Q.X. Exploring the interactive coercing relationship between urbanization and ecosystem service value in the Shanghai–Hangzhou Bay Metropolitan Region. J. Clean. Prod. 2020,253, 119803.

Figure 6. ESV of different land use types under different scenarios

Comment 15:L454-461: Seven lines and a half for one, very long sentence, which intertwines so much that it is not clear any more what the subject is. Please rephrase.

Response: Thank you for your comments. We have rephrased this sentence. Change to " Land use change is a complex dynamic process, based on an analysis of the present situation of land use in the past cannot fully express the qualitative change of land use types [73], more the lack of considering the impact of multiple development goals. Through analyzing the past and simulating future land use patterns under different scenarios, the study discusses the impact of future land use structure change on ESV under the influence of different development goals. The results enrich the research on the impact of land use change in the YRB on future ESV." (corresponding to lines 338-341 of the revised document). The details are as follows:

Line453-459: “Land use change is a complex dynamic process, based on an analysis of the present situation of land use in the past cannot fully express the qualitative change of land use types [73], more the lack of considering the impact of multiple development goals. Through analyzing the past and simulating future land use patterns under different scenarios, the study discusses the impact of future land use structure change on ESV under the influence of different development goals. The results enrich the research on the impact of land use change in the YRB on future ESV.”

Comment 16:L486: Could you please elaborate, what would a microscopic perspective be in the context of the Yellow River Basin?

Response: Thank you for your question. The micro perspective refers to the single ecosystem, the types of ecosystem services involved, and the smaller research scale. In future studies, we can discuss and analyze the impacts of single ecosystems such as grassland and forest land, and the internal tradeoffs and synergies among ecosystem service types included in supply and regulation services, as well as human well-being and urbanization development. In addition, higher resolution land use data can also be used to explore the supply-demand relationship of ecosystem service value at a finer scale, such as city or county in the Yellow River Basin. In the future, we will further collect and sort out land use and ecosystem data, and study the spatial response of ESV in the watershed by establishing a grid, so as to further improve the analytical accuracy of ecosystem service value measurement.

We have restated this sentence in the article according to your suggestion. “To better understand the correlation between land use and ESV and maximize watershed ESV, we will make further efforts in future research to integrate land use and ecosystem data, and study the spatial response of watershed ESV by establishing a grid, so as to further improve the analytical accuracy of ecosystem service value measurement.”(corresponding to lines 481-486 of the revised document). The details are as follows:

Line481-486:

“To better understand the correlation between land use and ESV and maximize watershed ESV. We will make further efforts in future research to integrate land use and ecosystem data, and study the spatial response of watershed ESV by establishing a grid, so as to further improve the analytical accuracy of ESV measurement.”
